# Combined Intermediate Cervical Plexus and Costoclavicular Block for Arthroscopic Shoulder Surgery: A Prospective Feasibility Study

**DOI:** 10.3390/jpm13071080

**Published:** 2023-06-29

**Authors:** Jeong Uk Han, Chunwoo Yang, Jang-Ho Song, Jisung Park, Hyeonju Choo, Taeil Lee

**Affiliations:** Department of Anesthesiology and Pain Medicine, School of Medicine, Inha University, Incheon 22212, Republic of Korea; jwhanan@nate.com (J.U.H.); snowguy@naver.com (J.-H.S.); zzangwltjd@naver.com (J.P.); guswncn@naver.com (H.C.); prupruda@naver.com (T.L.)

**Keywords:** anesthesia, brachial plexus, cervical plexus, nerve block, shoulder surgery

## Abstract

A combined cervical plexus and costoclavicular block provides effective shoulder analgesia without the risk of hemidiaphragmatic paralysis. However, whether this technique can also provide effective anesthesia for shoulder surgery remains unknown. Therefore, this study aimed to assess the feasibility and adverse effects of combined blocks in arthroscopic shoulder surgery. Fifty patients scheduled for arthroscopic shoulder surgery were prospectively enrolled. Intermediate cervical plexus (5 mL of 0.5% ropivacaine) and costoclavicular (20 mL of 0.5% ropivacaine) blocks were administered under ultrasound guidance. The block procedure time, needle pass, patient discomfort, anesthesia quality, onset time, postoperative analgesia quality, adverse events, and patient satisfaction were assessed. Surgical and block success were achieved in 45 (90%; 95% confidence interval [CI], 78–97%) and 44 (88%; 95% CI, 76–95%) patients, respectively. Three patients required local anesthetic supplementation, and two required general anesthesia. The incidence of hemidiaphragmatic paralysis was 12.0% (95% CI, 4.5–24.3%). Postoperative pain control was effective for the first 24 h postoperative. Neurological deficits were not observed. The patients reported a high level of satisfaction. This study revealed that a combined cervical plexus and costoclavicular block provided effective surgical anesthesia for arthroscopic shoulder surgery with a 12% incidence of hemidiaphragmatic paralysis. Further randomized studies comparing this technique with interscalene block are required.

## 1. Introduction

The interscalene brachial plexus block (ISB) is one of the most effective anesthetic and analgesic techniques for shoulder surgery. Despite its efficacy, one of the disadvantages of this technique is the occurrence of ipsilateral hemidiaphragmatic paralysis (HDP) owing to its close anatomical relationship with the phrenic nerve. Consequently, it poses a great risk for patients who are obese or have impaired respiratory function [1].

Several diaphragm-sparing alternatives to ISB, including upper trunk block [2,3], supraclavicular brachial plexus block [4], anterior suprascapular nerve block [5], combined infraclavicular brachial plexus block, and suprascapular nerve block [6], have recently been introduced and investigated for pain management following shoulder surgery [7]; however, these studies mainly focused on shoulder analgesia.

Costoclavicular brachial plexus block (CCB) involves anesthesia of the brachial plexus in the costoclavicular space at the level of its cords [8]. An anatomical study revealed that injection of a dye in the costoclavicular space stained all the trunks, cords, and divisions of the brachial plexus, as well as the suprascapular nerve [9]. Moreover, Aliste et al. reported that CCB (20 mL of 0.5% levobupivacaine and 5 g/mL of epinephrine) combined with an intermediate cervical plexus block (CPB) provides analgesia comparable to that of ISB for shoulder surgery without the risk of HDP [10]. These findings suggest that a combination of intermediate CPB and CCB could provide surgical anesthesia for shoulder surgery, avoiding the risk of HDP. However, limited information is available regarding this topic [7].

The present study aimed to evaluate the feasibility, efficacy, and complications of a combined block for regional anesthesia during arthroscopic shoulder surgery. We hypothesized that a combination of intermediate CPB and CCB would provide effective anesthesia and analgesia for arthroscopic shoulder surgery without the risk of HDP. The primary outcome included surgical success rate, whereas the secondary outcomes included block-related outcomes (block performance time, number of needle passes, onset time, and patient discomfort), block success rate, anesthesia quality, diaphragmatic excursion, incidence of HDP, postoperative analgesia quality, patient satisfaction, and adverse events.

## 2. Materials and Methods

### 2.1. Ethics

This prospective feasibility study was approved by the Institutional Review Board of Inha University Hospital (#2021-05-004-001), and written informed consent was obtained from all participants. This study was conducted in accordance with the principles of the Declaration of Helsinki. The trial was registered before patient enrollment in the Clinical Trial Registry of Korea (KCT0006297, Principal investigator: C Yang, https://cris.nih.go.kr/, accessed on 6 August 2021). To prepare this report, we adhered to the Consolidated Standards of Reporting Trials (CONSORT) extension guidelines for pilot and feasibility trials.

### 2.2. Participants

After obtaining written informed consent, patients scheduled for elective arthroscopic shoulder surgery between June and November 2021 were prospectively enrolled. The inclusion criteria were age between 19 and 80 years and American Society of Anesthesiologists (ASA) physical status classes I–III. The exclusion criteria were age <19 years, body mass index >35 kg/m^2^, patient refusal, pre-existing neurological deficits or neuropathy affecting the operative limb, severe respiratory disease, mental or psychiatric disorders preventing assessment, coagulation disorders, severe renal or hepatic failure, pregnancy, infection near the block procedure area, and known allergies to local anesthetics. Patient demographics, including age, sex, weight, height, body mass index, and ASA class, were recorded.

### 2.3. Block Technique

On the day of the surgery, patients arrived in the block room 1 h before the scheduled surgery. Standard ASA monitoring, including pulse oximetry, noninvasive blood pressure measurements, and electrocardiography, was performed. All blocks were performed by the same anesthesiologist (CY), with experience in both techniques. Intravenous (IV) midazolam was administered for anxiolysis. An ultrasound device (Viamo c100; Canon Medical Systems Co., Otawara, Japan) with a high-frequency (5–13 MHz) linear transducer was used in all cases. A pre-scan was performed to optimize the ultrasound apparatus settings. Sterile preparations included scrubbing the skin area with a combination of 2% chlorhexidine and 70% isopropyl alcohol preparation solution (HEXITANOL solution, Firson, Cheonan, Republic of Korea), sterile transducer covers (Ultrasound Probe Cover, Yafho Bio-Technology Co., Ltd, Guangzhou, China), and sterile ultrasound gel (SUPERSONIC, SUNGHEUNG Corporation, Bucheon, Republic of Korea). A 50 or 80 mm insulated needle (UniPlex NanoLine; Pajunk, Geisingen, Germany) and nerve stimulator (Stimuplex HNS 12; B. Braun, Melsungen, Germany) were used for all blocks.

CPB was performed first followed by CCB. First, intermediate CPB was performed under ultrasound guidance without a nerve stimulator since the cervical plexus is a sensory neural plexus. The patients were placed in a supine position with arm adducted, and the head was turned to the contralateral side. A linear transducer was placed at the posterolateral aspect of the neck in transverse orientation. The needle was inserted at the midpoint of the posterior border of the sternocleidomastoid muscle. After administration of 1 to 2 mL of 2% lidocaine to anesthetize the skin using a lateral-to-medial in-plane technique, a 5 cm block needle was positioned in the intermuscular plane between the sternocleidomastoid and scalene muscles (Figure 1A) [11], which often provided visualization of the superficial cervical plexus. After negative blood aspiration, 5 mL of 0.5% ropivacaine was slowly injected.

Subsequently, CCB was performed. Patients were placed in a supine position with the ipsilateral arm abducted. Initially, the transducer was placed directly over the middle third of the clavicle in the transverse orientation. The transducer was moved caudally and placed transversely below the middle third of the clavicle. The axillary artery was identified beneath the subclavian muscle in the costoclavicular space. The brachial plexus was identified as a compact group of nerves located lateral to the axillary artery below the clavicle. After sterilization with an antiseptic solution, a stimulating needle was advanced using an in-plane approach from the lateral to medial direction and positioned at the center of the brachial plexus cords (Figure 1B) [12]. The nerve stimulators were set at 0.5 mA, 0.1 ms, and 1 Hz. The needle position was adjusted when the motor response decreased below a stimulation of <0.3 mA. Following a negative blood aspiration, 20 mL of 0.5% ropivacaine was slowly injected in increments.

### 2.4. Preoperative Evaluation

The block procedure time, number of needle passes, patient discomfort, and adverse events (vascular puncture, paresthesia, hoarseness, Horner’s syndrome, and local anesthetic systemic toxicity) were recorded by an anesthesia nurse who assisted the anesthesiologist in performing the block. Block performance time was defined as the time between the initial needle insertion and final needle withdrawal during the entire block placement. The initial needle insertion was counted as the first pass, whereas any subsequent needle advancement preceded by a retraction of at least 10 mm was considered an additional pass. Patient discomfort during block placement was assessed using a numerical rating scale (NRS) of 0 to 10 (0 = no pain; 10 = worst pain imaginable).

Sensory and motor blocks were assessed by an anesthesiologist every 5 min for up to 30 min after injection. Sensory block was assessed at the level of the C4–C6 dermatome (C4, shoulder tip; C5, skin over the distal deltoid muscle; C6, lateral aspect of the forearm and thumb) relative to the contralateral arm using a three-point scale with a pin-prick test as follows: 0 for normal sensation, 1 for loss of sensation to pinprick, and 2 for complete loss of sensation to touch. Motor blockade of the four nerves (axillary, suprascapular, subscapular, and lateral pectoral nerve) was evaluated using a handheld electromechanical dynamometer (Echo MMT; JTECH Medical Industries Inc., Midvale, UT, USA). Baseline muscle power was measured before block placement. Motor function was assessed using the following forces: shoulder abduction (axillary nerve), shoulder external rotation (suprascapular nerve), shoulder internal rotation (subscapular nerve), and humeral internal rotation (lateral pectoral nerve) with maximum voluntary isometric contraction. Shoulder abduction measurements were performed in the sitting position, whereas the remaining measurements were performed in the supine position.

The block onset time was defined as ≥1 sensory nerve dermatome and a 50% decrease in the motor block of the four nerves. Block success was defined as the absence of pinprick sensation in the C4–C6 dermatome and motor block reduction of ≥50% in the four nerves within 30 min. Block duration was evaluated in patients who reported no pain while in the post-anesthesia care unit and was defined as the time between the completion of the local anesthetic injection and the first pain sensation experienced in the affected extremity.

Diaphragmatic excursion was assessed before and 30 min after the administration of nerve block. With the patient in the supine position, a curvilinear ultrasound transducer (Viamo c100, 1–5 MHz) was used to perform the scan using a low intercostal or subcostal approach, with the liver on the right or spleen on the left as an acoustic window. Diaphragmatic excursion was measured during deep inspiration and sniffing tests by using M-mode ultrasonography. During the “sniff test”, diaphragmatic movement was evaluated from the expiratory position during quick nasal inspirations of air. The normal caudal motion of the diaphragm with inspiration was designated as positive (+), while the cephalad paradoxical motion was considered negative (−). Each measurement was performed twice and the values were averaged. Diaphragmatic movement was measured in centimeters. A reduction in ipsilateral hemidiaphragmatic excursion of >75% relative to the baseline, absence of hemidiaphragmatic movement, or paradoxical movement was defined as complete HDP. A reduction in hemidiaphragmatic excursion between 25% and 75% was defined as partial HDP, and a reduction in excursion of <25% was defined as no HDP. HDP incidence was defined as the number of complete and partial HDP cases.

### 2.5. Perioperative Management

Following the block assessment, the patients were transferred to the operating room. At the first incision, the attending anesthesiologist assessed block adequacy. Surgical success was defined as the absence of any requirement for supplemental intravenous analgesics, local anesthetic infiltration by the surgeon, or conversion to general anesthesia to complete the surgery. Block failure was identified as the absence of such factors. It was managed with local anesthetic infiltration, intravenous opioids, or conversion to general anesthesia, at the discretion of the attending anesthesiologist. After assessment of block adequacy, sedation was maintained with continuous intravenous propofol infusion to maintain the bispectral index in the range of 60–70 as required. Throughout the procedure, supplemental oxygen was administered via a mask at a rate of 6 L/min. At the end of the operation, the surgeon infiltrated all the portal sites with 3 mL of 0.2% ropivacaine. All patients received 0.3 mg of ramosetron for antiemetic prophylaxis.

Postoperative analgesia was standardized, with all the patients receiving 1 g of intravenous acetaminophen every 6 h and 30 mg of ketorolac every 12 h, regardless of pain status. Additionally, all patients received intravenous patient-controlled analgesia (PCA) consisting of fentanyl (bolus, 10 μg; lockout interval, 10 min). If additional pain management was required, 25 mg of intravenous meperidine was administered. The opioid requirement was converted to fentanyl equivalents (30 mg of meperidine equivalent to 100 μg of fentanyl) and added to the PCA regimen.

Pain scores were assessed by an investigator using the NRS at 0.5, 3, 6, and 24 h post operation. The use of intravenous fentanyl for PCA was recorded at the same time points. At 0.5 and 24 h post operation, the investigator inquired whether the patient had experienced dyspnea or postoperative nausea and vomiting. Patient satisfaction was measured on a 10-point scale (0 = very dissatisfied to 10 = very satisfied) 24 h post operation. They were evaluated by the attending surgeon for block-related complications, including paresthesia and/or motor deficits, approximately four weeks after surgery.

### 2.6. Sample Size Calculation and Statistical Analysis

The sample size was calculated based on a previous study recommending a sample size of at least 50 participants per group [13]. Descriptive statistics were used for statistical analysis. Data were presented as mean/standard deviation (SD) or median (interquartile range). Categorical data were presented as numbers (percentages). SPSS software (version 18.0; SPSS Inc., Chicago, IL, USA) was used for data computation and statistical analyses. A *p* value of ˂0.05 was considered statistically significant.

## 3. Results

Fifty-five consecutive patients scheduled for arthroscopic shoulder surgery were assessed for eligibility, of which five were excluded because they refused to participate in the study. Finally, 50 patients completed the study and were included in the analysis. The patient demographics and surgical data are shown in Table 1.

Surgical and block success were achieved in 45 (90%; 95% confidence interval [CI], 78–97%) and 44 (88%; 95% CI, 76–95%) patients, respectively. Three patients reported incisional pain at the posterior portal site due to incomplete sensory blockade of the supraclavicular nerve and required intravenous fentanyl (50–100 μg mL) for rescue analgesia and local anesthetic infiltration (2% lidocaine, 3–4 mL). Two patients required general anesthesia to proceed with the surgery because of incomplete blockade of the brachial plexus; one patient showed incomplete blockade of the subscapular nerve, while the other showed incomplete blockade of both the subscapular and axillary nerves.

Figure 2 shows the progression of sensory and motor block in different areas over 30 min. The block-related outcomes are shown in Table 2. Before the block administration, the ultrasound examination of the diaphragmatic excursions revealed positive motion in all patients, and the baseline diaphragmatic excursion was 4.9 (1.0) cm. At 30 min following the block administration, the diaphragmatic excursion was 4.5 (1.8) cm. No significant differences were observed in terms of outcomes (*p* = 0.179). The incidence of HDP was 12.0% (95% CI, 4.5–24.3%), and all HDP cases were classified as complete. None of the patients experienced dyspnea.

Postoperative pain control was effective during the first 24 h post operation (Table 3). No neurological deficits were observed during the follow-up. Patients reported a high level of satisfaction (score = 10 for all).

## 4. Discussion

The present study aimed to investigate the feasibility and efficacy of combined intermediate CPB and CCB for providing surgical anesthesia in patients undergoing arthroscopic shoulder surgery. This study demonstrated that a combination of intermediate CPB and CCB can provide adequate surgical anesthesia and analgesia for arthroscopic shoulder surgery. The incidence of HDP was 12.0% (95% CI, 4.5–24.3%), and the combined block provided a high degree of patient satisfaction. No severe adverse events were reported.

Procedures in the shoulder joint required the blockade of the supraclavicular (from the cervical plexus), axillary, suprascapular, lateral pectoral, and subscapular (from the brachial plexus) nerves. Therefore, it is necessary to block both the cervical and brachial plexuses. Theoretically, a combination of intermediate CPB and CCB can provide adequate anesthesia for shoulder surgery. In this study, the incidence of surgical anesthesia was achieved in 45 (90%; 95% CI, 78–97%) patients, while 5 (10%) of them required supplemental analgesia. This suggested that this anesthetic technique provides a reliable blockade of the relevant nerves for arthroscopic shoulder surgery, which is consistent with the findings of a previous anatomical study [9]. Musso et al. reported that a triple combination of cervical, suprascapular, and lateral sagittal infraclavicular plexus blocks provides acceptable anesthesia for arthroscopic shoulder surgery [14]. In contrast, our block technique required only two separate blocks because of the spread of the local anesthetic into the suprascapular nerve following CCB. Theoretically, this could reduce the procedural time, number of needle passes, and patient discomfort. In addition, this combined block can reduce the local anesthetic dose required for a successful block.

The choice of local anesthetics in this study is worth noting. One could argue that a rapid onset is essential for surgical anesthesia. Thus, selecting 0.5% ropivacaine as a local anesthetic might not be considered appropriate for surgeries requiring rapid anesthesia. However, shoulder surgery generally results in moderate to severe postoperative pain. To prolong the duration of postoperative analgesia, we selected 0.5% ropivacaine as a local anesthetic for the combined block. In this study, the block onset time was 15 min (10–20 min), whereas a previous dose-finding study [15] reported a median onset time of 30 min (20–45 min) with CCB only for forearm or hand surgery. However, our data cannot be compared to those of other studies due to the differences in the criteria used to determine the onset time of the block. The utilization of a double-injection technique for CCB may improve the block onset time [16].

One of the major advantages of this combined block is a 0% incidence of HDP. Aliste et al. reported that this combined block technique provides similar shoulder analgesia without the risk of HDP [10]. Unfortunately, in our study, the incidence of HDP was 12.0% (95% CI, 4.5–4.3%) despite the use of the same volume (20 mL) of local anesthetic. Several factors may be associated with transient side effects of the combined block. First, due to the relatively low mean body mass index of our cohort, the local anesthetic dose (20 mL) could have directly spread to the phrenic nerve following CCB. Another plausible explanation is the spread of the local anesthetic through the prevertebral fascia into the phrenic nerve following intermediate CPB. The permeability of the prevertebral fascia to local anesthetics is not well established, and a previous study [17] reported that the superficial cervical space communicates with the deep cervical space, resulting in the blockade of the phrenic nerve. In contrast, another anatomical study demonstrated that the prevertebral fascia is impermeable to the injected dye solution, suggesting the absence of HDP following intermediate CPB [18,19].

Another contributing factor was the presence of anatomical variations. The accessory phrenic nerve is a relatively common anatomical variant, with an incidence of 36.5% [20]. The accessory phrenic nerve arises from the ansa cervicalis, nerve to the subclavian nerve, and the C5–C6 nerve roots [20]. Thus, HDP can occur with both CPB and CCB, suggesting that the combined block cannot circumvent these adverse events, thus limiting its use in patients with severe respiratory diseases.

This study had several limitations. First, this was not a randomized controlled study. Thus, controlled trials are needed to compare this technique with standard regional anesthetic techniques, such as ISB, and to evaluate their efficacy. Second, the patient cohort in our study had relatively low body mass index. Thus, our results may not apply to patients suffering from obesity. Third, motor block was assessed using a force dynamometer, which may limit the generalizability of the definition of block success. Therefore, a simpler motor-block assessment method is desirable. Fourth, we did not assess the imaging time; in our experience, the time required was less than 1 min. Fifth, this study only included arthroscopic shoulder surgeries. Further studies, including various surgical procedures, are warranted to confirm these results. Finally, based on a previous study [10], we used 5 and 20 mL of local anesthetic for CPB and CCB, respectively. Thus, a dose-finding study is warranted for CCB used in shoulder anesthesia.

## 5. Conclusions

This study demonstrated the feasibility and efficacy of combining intermediate CCB and CPB for arthroscopic shoulder surgery. Combined CPB and CCB can be an alternative anesthetic technique to ISB, offering a reduced risk of HDP in arthroscopic shoulder surgery. Further randomized studies comparing this block with ISB are required.

## Figures and Tables

**Figure 1 jpm-13-01080-f001:**
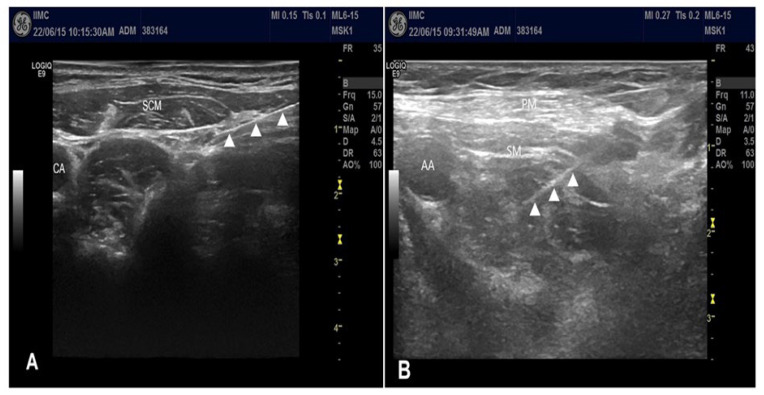
(**A**) Ultrasound-guided intermediate cervical plexus block; (**B**) ultrasound-guided costoclavicular brachial plexus block. ▲ indicates needle shaft. AA, axillary artery; CA, carotid artery; PM, pectoralis major muscle; SCM, sternocleidomastoid muscle; SM, subclavius muscle.

**Figure 2 jpm-13-01080-f002:**
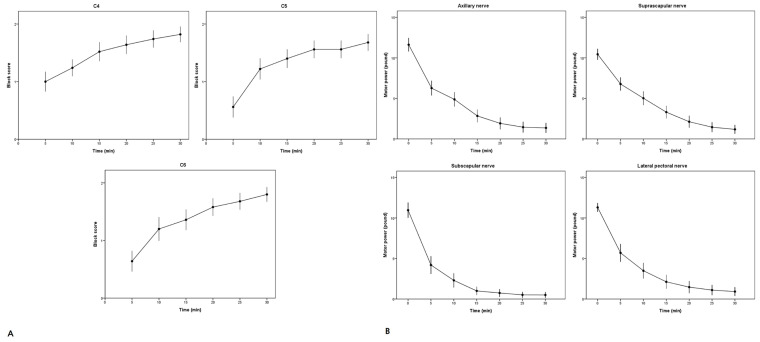
(**A**) Evolution of sensory block quality in the territories of the C4, C5, and C6 dermatomes; (**B**) evolution of the motor block quality in the territories of the axillary, suprascapular, subscapular, and lateral pectoral nerves.

**Table 1 jpm-13-01080-t001:** Patient demographics and surgical data.

Age, years	62 (9)
Sex, male/female, *n*	21/29
Weight, kg	65 (10)
Height, cm	160 (9)
BMI, kg/m^2^	25.1 (2.8)
ASA physical status, I/II/ΙΙΙ, *n*	17/32/1
Duration of surgery, min	116 (26)
Surgical procedure, *n*	
Rotator cuff repair	46
Capsulotomy	3
Other ligament repair	1

Values are expressed as mean (standard deviation) or number of patients. ASA, American Society of Anesthesiologists; BMI, body mass index.

**Table 2 jpm-13-01080-t002:** Block-related outcomes.

Procedure time, min	3.7 (0.9)
Number of needle passes, 0–10 scale	2 (2–3)
Patient discomfort, 0–10 scale	3 (2–3)
Block onset time, min	15 (10–20)
Block duration, h	13.6 (4.9)
Paresthesia	5 (10)
Vascular puncture	0 (0)
Horner’s syndrome	0 (0)
Hoarseness	3 (6)

Values are expressed as mean (standard deviation), median (interquartile range), or number (%).

**Table 3 jpm-13-01080-t003:** Postoperative outcomes.

Pain Score	
0.5 h	0 (0–0)
3 h	0 (0–0)
6 h	0 (0–3)
24 h	3 (0–4)
Opioid (fentanyl) consumption, µg	
0.5 h	0 (0–0)
3 h	0 (0–15)
6 h	15 (0–45)
24 h	135 (60–293)
Postoperative nausea and vomiting	
0.5 h	0
24 h	0
Dyspnea	
0.5 h	0
24 h	0

Values are expressed as median (interquartile range) or number.

## Data Availability

The datasets are available from the corresponding author upon reasonable request.

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
