# Peer review of "Combined Intermediate Cervical Plexus and Costoclavicular Block for Arthroscopic Shoulder Surgery: A Prospective Feasibility Study"

_jpm, 2023, doi:10.3390/jpm13071080_

Round 1

Reviewer 1 Report

It is a good attempt to find alternate diaphragm sparing options for regional blocks for shoulder procedures. Even in this under powered study, the incidence of HDP is 12%. Definitely worth exploring, though!

As the authors have noted, it is a feasibility study - needs a further RCT to test the validity of this combination of blocks.

Would like to know the cause for 10% failure rate of this block. How did these patients report their pain experience post operatively? You report that all of the patients expressed complete satisfaction.

P1L24: 'The patient reported...' -should read 'Patients reported...'

Author Response

Dear reviewer.

We agree with your recommendation that there is a need for a randomized study to test the validity of this combined block. We are currently performing this trial.

  1. regarding block failure -> In this study, the surgical success was achieved in 45 (90%) patients.  Three were incomplete blockade of the cervical plexus, two were incomplete blockade of the brachial plexus. Despite block failure, three patients with incomplete cercial plexus block did not reported any postoperative pain in the recovery room. Two patients with incompete brachial plexus block reported mild pain (VAS 3-4), but they did not require rescue analgesics in the PACU, We think that this was associated with the use of ropivacaine in our study. This long-acting agent has a longer onset time compared to intermediate local anesthetic. 

2. P1L24 "The patient reported ..." Patient reported.

-> Thanks for your suggestion. We corrected it as you suggested.

Thanks for your kind review.

From C Yang

Reviewer 2 Report

This is a non-randomized study and was not conducted by double blinding. Additionally, it lacks originality. The authors mention in the discussion that Aliste et al reported a similar study and used a rigorous methodology finding similar clinical results. It is recommended that the authors use randomization and double-blind techniques in their future studies, which include a control group that serves as a reference to determine the aesthetic and/or analgesic efficacy of the techniques under evaluation. The above would increase the rigor of your study.

Author Response

Dear reviwer.

In this prospective feasibility study, we focused on anesthesia without the risk of hemidiaphragmatic paralysis, but not analgesia without the risk of diaphragmatic paralysis for shoulder surgery.

We agree with your excellent recommendations. We have performed another randomized double-binded study that compares interscalene block with combined cervical plexus block and costoclavicular block for shoulder anesthesia. 

Thanks for your excellent review.

From C Yang

Reviewer 3 Report

This is a well-written manuscript describing the use of cervical plexus and costoclavicular blocks for shoulder surgery.

The authors discuss the the blocks can be placed in a relatively short period of time.  However, their patients were BMI about 25 that is far less than what most of us encounter in clinical practice.  The authors should briefly mention this. 

Author Response

Dear reviewer.

We add your suggestion at the limitation of discussion in the manuscript.

Thanks for your suggestion.

Round 2

Reviewer 2 Report

OK